**Data Availability Statement:** https://wwwn.cdc.gov/nchs/nhanes/Default.aspx.

**Funding:** National Institute on Minority Health and Health Disparities, 2U54MD007601-36, Assistant

# Exploring the diagnostic performance of machine learning in prediction of metabolic phenotypes focusing on thyroid function

**Hyeong Jun Ahn**[1], **Kyle Ishikawa**[1]*, **Min-Hee Kim**[2]*

**1** Department of Quantitative Health Sciences, John A. Burns School of Medicine, University of Hawaii at Manoa, Honolulu, Hawaii, United States of America, **2** Division of Endocrinology and Metabolism, Department of Internal Medicine, Eunpyeong St. Mary's Hospital, College of Medicine, The Catholic University of Korea Seoul, South Korea

* kylemish@hawaii.edu (KI); mhkim6@hawaii.edu (MHK)

## Abstract

In this study, we employed various machine learning models to predict metabolic phenotypes, focusing on thyroid function, using a dataset from the National Health and Nutrition Examination Survey (NHANES) from 2007 to 2012. Our analysis utilized laboratory parameters relevant to thyroid function or metabolic dysregulation in addition to demographic features, aiming to uncover potential associations between thyroid function and metabolic phenotypes by various machine learning methods. Multinomial Logistic Regression performed best to identify the relationship between thyroid function and metabolic phenotypes, achieving an area under receiver operating characteristic curve (AUROC) of 0.818, followed closely by Neural Network (AUROC: 0.814). Following the above, the performance of Random Forest, Boosted Trees, and K Nearest Neighbors was inferior to the first two methods (AUROC 0.811, 0.811, and 0.786, respectively). In Random Forest, homeostatic model assessment for insulin resistance, serum uric acid, serum albumin, gamma glutamyl transferase, and triiodothyronine/thyroxine ratio were positioned in the upper ranks of variable importance. These results highlight the potential of machine learning in understanding complex relationships in health data. However, it's important to note that model performance may vary depending on data characteristics and specific requirements. Furthermore, we emphasize the significance of accounting for sampling weights in complex survey data analysis and the potential benefits of incorporating additional variables to enhance model accuracy and insights. Future research can explore advanced methodologies combining machine learning, sample weights, and expanded variable sets to further advance survey data analysis.

## Introduction

Obesity is rapidly increasing worldwide, imposing a significant burden on health systems [1] and is associated with various adverse health outcomes especially in terms of dysregulated metabolism [2]. Traditionally and practically, obesity is defined by a single dimension, body

Professor Hyeong Jun Ahn National Institute of General Medical Sciences, U54GM138062, Assistant Professor Hyeong Jun Ahn.

**Competing interests:** The authors have declared that no competing interests exist.

mass index (BMI). However, considering heterogeneous clinical presentations of obesity, such as the existence of metabolically healthy individuals within the obese population, a more sophisticated way of obesity classification is needed for understanding of the mechanistical association between obesity and adverse health outcomes as well as for proper management of obese population. Based on both BMI and metabolic presentations, metabolic phenotypes are classified as metabolic healthy normal weight (MHNW), metabolic healthy obesity (MHO), metabolic unhealthy normal weight (MUNW), and metabolic unhealthy obesity (MUO). Individuals with metabolically healthy obesity are known to exhibit various favorable characteristics in terms of metabolism [3], despite being obese, and have a relatively lower cardiovascular risk compared to the metabolically unhealthy population [4]. However, the clinical significance of MHO, including the existence of specific management options, and the underlying mechanistic distinctions between MUO and metabolically unhealthy normal weight (MUNW) are not fully comprehended.

As thyroid hormone is a main regulator of energy metabolism [5], altered glucose [6, 7] and lipid [8, 9] metabolism are observed in overt thyroid diseases. Even in subtle changes of thyroid function such as subclinical hypothyroidism and hyperthyroidism, those metabolic changes are found [10, 11]. Furthermore, variations of thyroid function within normal reference ranges were suggested to be related to metabolic derangement [12–15] and blood pressure [16]. In this context, several studies evaluated and found the association of thyroid function within reference ranges and metabolic phenotypes [17–20]. However, there is currently no consistent consensus, which may be attributed to variations in the variables used, different definitions of metabolic syndrome, and variations in the populations studied.

Machine learning (ML), a form of artificial intelligence, is an automated learning process that allows programs to solve problems by analyzing data, thus facilitating data comparison and classification [21]. By learning from and acting on data, it enables to generate an algorithm to predict a certain outcome. The accumulation of diverse and extensive healthcare data presents favorable conditions for the development of diagnostic and prognostic prediction models using machine learning techniques. Various ML methods have been applied in healthcare settings, particularly in the predictive analysis of conditions such as high blood pressure and diabetes [22]. Recently, Analyses have also been attempted to predict various diseases beyond hypertension and diabetes through the language model analysis of symptoms [23], further exemplifying how the domain of machine learning has expanded to encompass not only the analysis of skin lesions using imaging data [24] but also the prediction and management of physiological conditions [25], demonstrating a broadening scope of applications in healthcare. In the field of thyroid research, several studies have utilized machine learning techniques for investigation. In the early stages, there was study utilizing neural networks to predict thyroid function status based on laboratory and clinical data [26]. More recently, numerous research studies have been conducted using various machine learning techniques to predict the diagnosis of thyroid nodules [27]. Recent study explored the factors influencing TSH levels using commonly collected demographic information and laboratory parameters [28].

Given the lack of a consistent correlation between metabolic phenotypes and thyroid function, and with the underlying mechanisms still undisclosed, there is an opportunity to leverage artificial intelligence (AI) for the development of a predictive model for metabolic phenotypes. To date, limited research has explored the ability of machine learning methods to accurately predict metabolic phenotypes in conjunction with thyroid function. Thus, the aim of this study is to develop and evaluate the performance of machine learning methods for the prediction of metabolic phenotypes, with a special focus on thyroid function. Additional factors commonly associated with both thyroid function and metabolic phenotypes will be incorporated into the machine learning methods to create prediction models. Such an approach has

the potential to shed light on the complex relationship between these variables and advance our understanding in this domain.

Hyeong Jun Ahn, as the first author, spearheads the study's conceptualization, protocol development, and manuscript drafting, collaborating with co-authors to refine the submission. Additionally, Minhee Kim provides valuable clinical expertise through an exhaustive literature review, synthesizing existing knowledge to inform the study's background and rationale. Furthermore, Kyle Ishikawa plays a pivotal role in the research endeavor, spearheading all aspects of data management, organization, and analysis, ensuring the integrity and accuracy of the study's findings.

## Methods

Metabolic phenotypes, MHNW, MHO, MUNW, and MUO, were classified according to harmonized definition [29]. A cutoff value of BMI 30 was applied for definition of obesity, and unhealthy phenotypes were defined as having any of the following components: (1) fasting glucose $\geq$100 mg/dl (or taking medication/insulin for diabetes), (2) systolic blood pressure $\geq$130 mmHg or diastolic blood pressure $\geq$ 85 mmHg (or taking medication for hypertension), (3) triglyceride (TG) $\geq$150mg/dl, or (4) high density lipoprotein cholesterol (HDL-c) < 40 mg/dl in male, <50 mg/dl in female. We included demographic (gender, age, race, and smoking status) and thyroid laboratory test including free thyroxine, free triiodothyronine (FT3), total thyroxine (TT4), total triiodothyronine (TT3), thyrotropin (TSH), free T3/T4 ratio, total T3/T4, urine iodine creatinine ratio (UICR) and thyroid autoantibodies such as anti-thyroid peroxidase (TPO) antibody and anti-thyroglobulin (Tg) for the analysis. Laboratory variables that have been reported to be associated with metabolic dysregulation were also selected (ex. albumin, gamma glutamyl transferase, total bilirubin, uric acid, homeostatic model assessment for insulin resistance (HOMA-IR)) [30–33]. Additionally, we used other laboratory variables available in the dataset for ML prediction to improve the performance of ML, even though there have been few reports on their association with thyroid function or obesity.

For the continuous variables, we summarized the data using the median and interquartile range (IQR) (FT4 quartile (Q), FT3 Q, TT4 Q, TT3 Q, TSH Q, UICR Q, HOMA-IR Q). For the categorical variables, we presented the results as percentages and counts. We performed statistical tests, such as Fisher's exact test and Kruskal-Wallis rank sum test, to assess the significance of associations between the characteristics and metabolic phenotypes. The p-values indicated the statistical significance of these associations.

We employed supervised machine learning models, including random forest, boosted trees, logistic regression, lasso regression, K nearest neighbors, and a type of neural network called a multilayer perceptron to classify the metabolic phenotype. The dataset was divided into a 80% training set and a 20% testing set for model evaluation.

Preprocessing was applied to the data based on what was required or could improve model performance. Table 1 describes the preprocessing sets in the order that they were applied for each model. Only models or parameters calculated using the training set were applied to the testing set to avoid data leakage. For example, when imputing missing data in the testing set, the bagged tree models were created only using data from the training set. First, variables with zero variance were removed from the dataset since they do not contribute valuable information. Missing values were then imputed using bagged tree models, utilizing ensemble learning to estimate those values based on dataset relationships. Continuous variables with a correlation greater than 0.8 with another variable were removed to eliminate redundancy and mitigate multicollinearity. In the multinomial logistic regression, lasso regression, and neural network models, continuous variable skewness was reduced through the application of the Yeo-

**Table 1. Preprocessing steps.**

| Preprocessing Steps | Multinomial Logistic Regression | Lasso Regression | K Nearest Neighbors | Random Forest | Boosted Trees | Neural Network |
|---|---|---|---|---|---|---|
| Remove variables with zero variance | O | O | O | O | O | O |
| Impute missing values with bagged tree models | O | O | O | O | O | O |
| Remove continuous variables with correlation greater than 0.8 | O | O | O | O | O | O |
| Reduce continuous variable skewness with Yeo-Johnson transformation | O | O | O | X | X | O |
| Standardize continuous variables to mean of 0 and standard deviation of 1 | X | O | O | X | X | O |
| Create a *novel* level for categorical variables | O | O | O | O | O | O |
| Convert categorical variables to dummy variables | O | O | O | O | O | O |

O = preprocessing step implemented

X = preprocessing step not implemented

Johnson transformation, promoting a more symmetric distribution. Additionally, continuous variables in the lasso regression, K nearest neighbors, random forest, boosted trees, and neural network models were normalized to have a mean of 0 and a standard deviation of 1, ensuring balanced scales. To handle missing or unknown categorical values, a novel level was created for categorical variables in all models, preventing information loss. Finally, categorical variables were converted into dummy variables, generating binary indicators for each category, enhancing the models' handling of categorical data. By applying these preprocessing steps, we aimed to enhance the quality and suitability of the dataset for each respective machine learning model. An overview of the general ML steps is presented in Fig 1.

Hyperparameter tuning was conducted for each machine learning model, except for Linear Regression, using grid search and simulated annealing (SA). This technique randomly crawls the hyperparameter space to find the optimal combination. Simulated annealing will backtrack or restart its search when suboptimal combinations are found. To prevent overfitting, we employed a 5-fold cross-validation strategy, where the training set was divided into 5 smaller sets (k = 5). The model was trained on k-1 folds and validated on the remaining portion of the dataset. This process was repeated three times with different fold splits to eliminate bias from the initial split of the folds. The combination of hyperparameters that were selected were based on their performance in 15 total models (five folds times three repeats).

Subsequently, the tuned models were tested on the testing dataset, and the accuracy of the different models were assessed using metrics such as the Hand-Till area under the receiver operator curve (AUROC), accuracy, precision, recall, and F1-score. Confusion matrices were plotted to show predicted values compared to true values. Lastly, the top 20 important variables from the random forest model were plotted based on *purity*–the degree to which a node can separate the classes.

Table 2 presents the key parameters utilized for each classification model, outlining the specific settings employed for model training and evaluation.

## Lasso regression

Lasso regression performs feature selection by eliminating predictors that are redundant or do not decrease the prediction error. Complex relationships were created by adding every two- and three-way combination of interaction terms resulting in 19,600 new predictors. The tuned

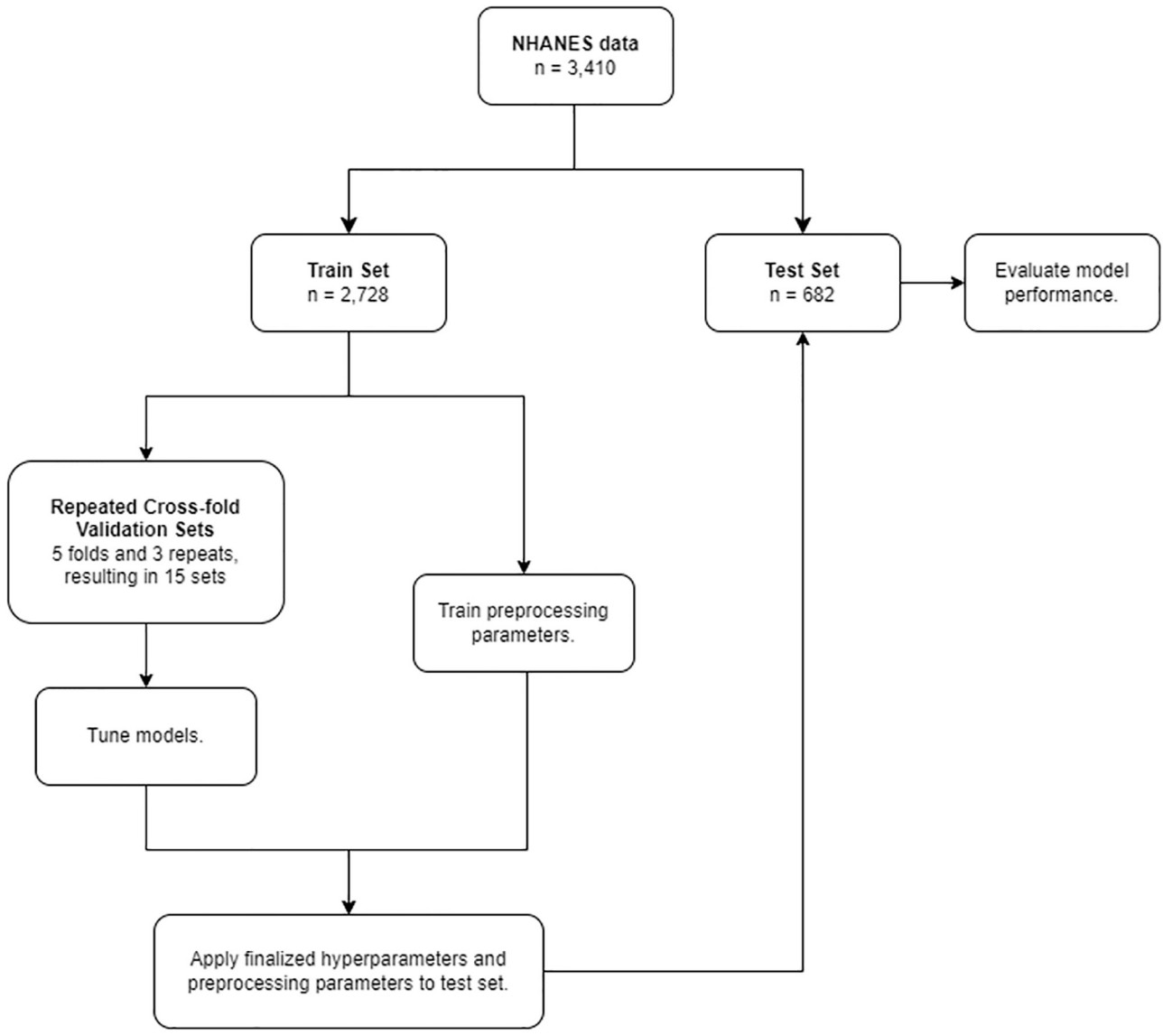

**Fig 1. Machine learning workflow.**

penalization parameter, $\lambda = 0.0193$, resulted in a model that only contained seven main effects and 206 interaction terms.

## K nearest neighbors

K nearest neighbors calculates distances from one observation to its surrounding observations, or *neighbors*. This model is good at predicting an outcome by examining neighbors that are the most similar to the input observation. The value of K was set to 101, indicating that the algorithm considers the 101 nearest neighbors when making predictions.

## Random forest

Random forest reduces the bias from individual decision trees that uses all of the training data. A RF is an ensemble of decision trees whose nodes are a random selection of predictors. This

**Table 2. Hyperparameter settings.**

| Model | Tuning Time | Parameter | Value |
|---|---|---|---|
| Multinomial Logistic Regression | NA | None | - |
| Lasso Regression | 10.31 minutes | Penalty | 0.0193 |
| K Nearest Neighbors | 6.97 minutes | K | 101 |
| Random Forest | 3.24 minutes | Number of trees | 490 |
| | | Number of predictors | 7 |
| | | Minimum node size | 45 |
| Boosted Trees | 2.74 hours | Number of trees | 1,695 |
| | | Number of predictors | 9 |
| | | Minimum node size | 33 |
| | | Tree depth | 7 |
| | | Learn rate | 0.00405 |
| Neural Network | 1.81 hours | Hidden units | 7 |
| | | Amount of regularization | 7.3 x 10^(-9) |
| | | Epochs | 404 |
| | | Learn rate | 0.296 |
| | | Activation function | ReLu |

reduces overfitting because the trees do not have access to all of the training data. Individually, they are weak models, but if many of them collected and vote on the outcome, they create a strong model. The number of trees in the forest was set to 490. Additionally, the algorithm uses 7 predictors for splitting nodes and requires a minimum node size of 45 observations.

### Boosted trees

Boosted trees is an extension of RF that sequentially improves on the error from the previous tree, hence, "boosting" the weak tree. In the end, the sequence of improved trees is combined to create an ensemble. Our implementation included 1,695 trees and each tree used 9 predictors for splitting nodes. The minimum node size was set to 33, and the tree depth was limited to 7. A learning rate of 0.00405 was used to control the contribution of each tree.

### Neural network

The neural network consisted of a single hidden layer with 7 units. This configuration is known as a multilayer perceptron and its hidden layer between its input and output layer is useful for modeling complex non-linear relationships. A visual of this neural network architecture has been presented in Fig 2. To prevent overfitting, an amount of regularization equal to 7.3 x 10^(-9) was applied. The training process spanned 404 epochs, and a learn rate of 0.296 was used. The activation function employed in the hidden layer was the rectified linear unit (ReLu).

## Results

### Baseline characteristics

A total of 3,4101 individuals were included in the analysis, categorized into five groups based on their characteristics: MHNW (N = 5901), MHO (N = 1211), MUNW (N = 1,6131), MUO (N = 1,0861), and unknown (N = 3). Significant differences were observed among these groups for various variables (Table 3).

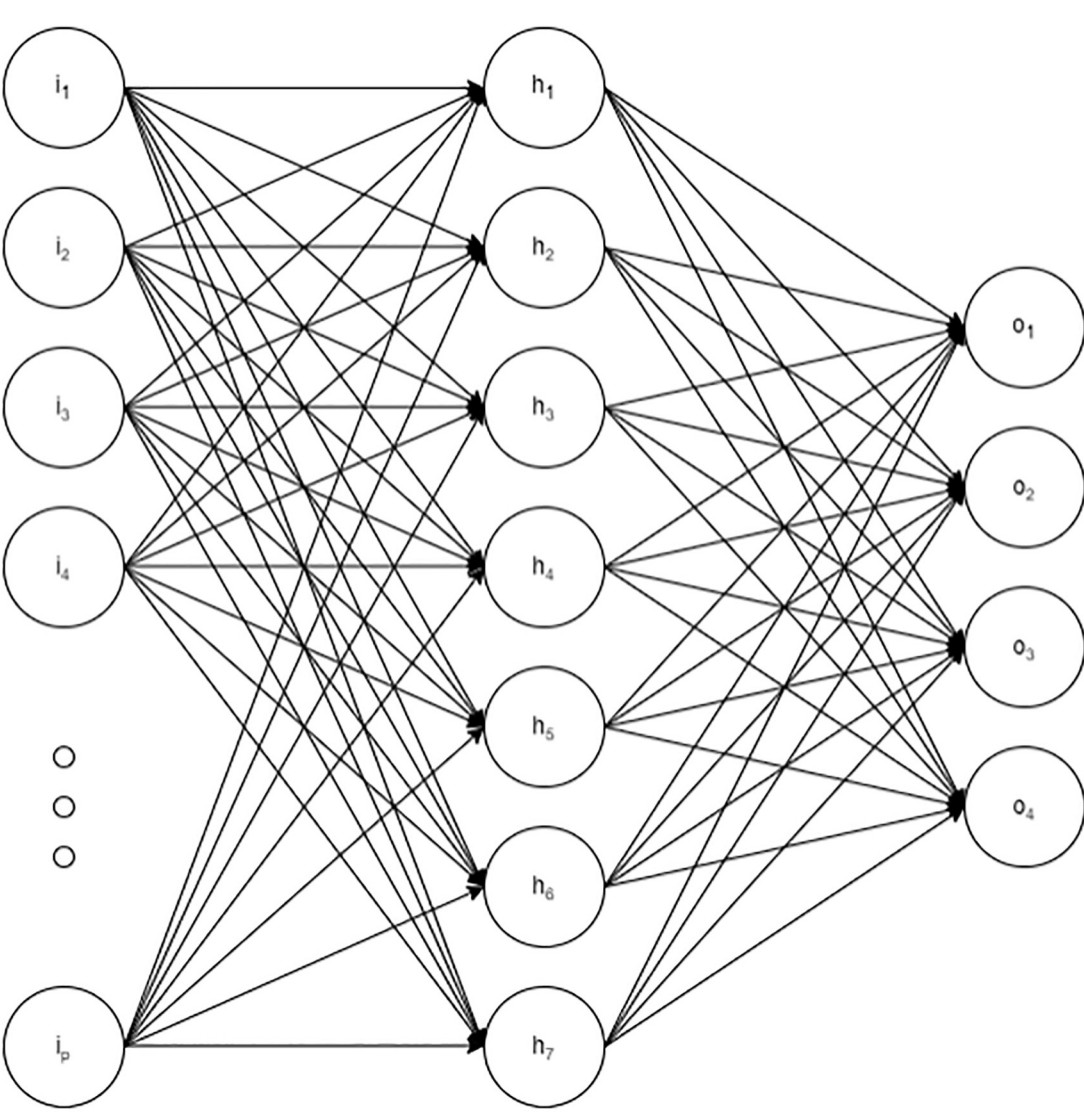

**Fig 2. Architecture of neural network.**

In terms of thyroid-related variables, the FT4Q quartiles did not show a statistically significant difference among the groups (p = 0.676). However, for the FT3Q quartiles, a significant difference was found (p < 0.001). The same pattern was observed for the TT4Q quartiles (p < 0.001) and the TT3Q quartiles (p < 0.001). The TSHQ quartiles also demonstrated a significant difference among the groups (p < 0.001). The UICRQ quartiles showed a significant difference (p < 0.001), as did the HOMAQ quartiles (p < 0.001).

Regarding demographic variables, a significant difference was observed for gender (p < 0.001), with a higher proportion of males in the MHNW and MHO groups compared to the MUNW, MUO, and unknown groups. Age (RIDAGEYR) also showed a significant difference among the groups (p < 0.001), with the MHNW group having a younger median age compared to the other groups.

**Table 3. NHANES dataset characteristics and comparison between phenotypes.**

| Characteristic | Overall | MHNW | MHO | MUNW | MUO | p-value[2] |
|---|---|---|---|---|---|---|
| | N = 3,410[1] | N = 590[1] | N = 121[1] | N = 1,613[1] | N = 1,086[1] | |
| FT4Q | | | | | | 0.676 |
| *Q1* | 1,103 (32%) | 183 (31%) | 41 (34%) | 539 (33%) | 340 (31%) | |
| *Q2* | 1,078 (32%) | 209 (35%) | 37 (31%) | 490 (30%) | 342 (31%) | |
| *Q3* | 737 (22%) | 120 (20%) | 25 (21%) | 345 (21%) | 247 (23%) | |
| *Q4* | 492 (14%) | 78 (13%) | 18 (15%) | 239 (15%) | 157 (14%) | |
| FT3Q | | | | | | <0.001 |
| *Q1* | 1,082 (32%) | 175 (30%) | 23 (19%) | 560 (35%) | 324 (30%) | |
| *Q2* | 742 (22%) | 142 (24%) | 25 (21%) | 328 (20%) | 247 (23%) | |
| *Q3* | 759 (22%) | 140 (24%) | 29 (24%) | 354 (22%) | 236 (22%) | |
| *Q4* | 824 (24%) | 133 (23%) | 44 (36%) | 370 (23%) | 277 (26%) | |
| *Unknown* | 3 | 0 | 0 | 1 | 2 | |
| TT4Q | | | | | | <0.001 |
| *Q1* | 759 (22%) | 174 (30%) | 26 (22%) | 376 (23%) | 183 (17%) | |
| *Q2* | 843 (25%) | 156 (27%) | 30 (25%) | 406 (25%) | 251 (23%) | |
| *Q3* | 873 (26%) | 127 (22%) | 26 (22%) | 428 (27%) | 292 (27%) | |
| *Q4* | 921 (27%) | 128 (22%) | 38 (32%) | 400 (25%) | 355 (33%) | |
| *Unknown* | 14 | 5 | 1 | 3 | 5 | |
| TT3Q | | | | | | <0.001 |
| *Q1* | 906 (27%) | 164 (28%) | 19 (16%) | 479 (30%) | 244 (22%) | |
| *Q2* | 867 (25%) | 173 (29%) | 27 (22%) | 392 (24%) | 275 (25%) | |
| *Q3* | 814 (24%) | 139 (24%) | 32 (26%) | 362 (22%) | 281 (26%) | |
| *Q4* | 822 (24%) | 114 (19%) | 43 (36%) | 379 (24%) | 286 (26%) | |
| *Unknown* | 1 | 0 | 0 | 1 | 0 | |
| TSHQ | | | | | | <0.001 |
| *Q1* | 932 (27%) | 192 (33%) | 46 (38%) | 447 (28%) | 247 (23%) | |
| *Q2* | 837 (25%) | 162 (27%) | 24 (20%) | 378 (23%) | 273 (25%) | |
| *Q3* | 796 (23%) | 114 (19%) | 33 (27%) | 402 (25%) | 247 (23%) | |
| *Q4* | 845 (25%) | 122 (21%) | 18 (15%) | 386 (24%) | 319 (29%) | |
| UICRQ | | | | | | <0.001 |
| *Q1* | 865 (26%) | 170 (29%) | 46 (38%) | 370 (24%) | 279 (26%) | |
| *Q2* | 789 (24%) | 152 (26%) | 35 (29%) | 341 (22%) | 261 (25%) | |
| *Q3* | 886 (27%) | 135 (23%) | 23 (19%) | 441 (28%) | 287 (27%) | |
| *Q4* | 787 (24%) | 124 (21%) | 17 (14%) | 411 (26%) | 235 (22%) | |
| *Unknown* | 83 | 9 | 0 | 50 | 24 | |
| HOMAQ | | | | | | <0.001 |
| *Q1* | 776 (23%) | 316 (54%) | 13 (11%) | 401 (25%) | 46 (4.3%) | |
| *Q2* | 856 (25%) | 183 (31%) | 42 (35%) | 483 (30%) | 148 (14%) | |
| *Q3* | 891 (26%) | 72 (12%) | 44 (37%) | 445 (28%) | 330 (31%) | |
| *Q4* | 861 (25%) | 13 (2.2%) | 21 (18%) | 274 (17%) | 553 (51%) | |
| *Unknown* | 26 | 6 | 1 | 10 | 9 | |
| RIAGENDR | | | | | | <0.001 |
| *Male* | 1,806 (53%) | 254 (43%) | 56 (46%) | 961 (60%) | 535 (49%) | |
| *Female* | 1,604 (47%) | 336 (57%) | 65 (54%) | 652 (40%) | 551 (51%) | |
| RIDAGEYR | 49 (35, 63) | 36 (26, 46) | 33 (26, 44) | 54 (39, 68) | 52 (38, 64) | <0.001 |
| smok_stat2 | | | | | | <0.001 |
| *Non Smoker* | 2,651 (78%) | 464 (79%) | 110 (91%) | 1,203 (75%) | 874 (81%) | |

*(Continued)*

**Table 3.** (Continued)

| Characteristic | Overall | MHNW | MHO | MUNW | MUO | p-value[2] |
|---|---|---|---|---|---|---|
| | N = 3,410[1] | N = 590[1] | N = 121[1] | N = 1,613[1] | N = 1,086[1] | |
| *Current Smoker* | 756 (22%) | 125 (21%) | 11 (9.1%) | 409 (25%) | 211 (19%) | |
| *Unknown* | 3 | 1 | 0 | 1 | 1 | |
| TPO_stat | | | | | | 0.928 |
| *Negative* | 3,109 (92%) | 534 (91%) | 112 (93%) | 1,473 (92%) | 990 (92%) | |
| *Positive* | 286 (8.4%) | 53 (9.0%) | 9 (7.4%) | 135 (8.4%) | 89 (8.2%) | |
| *Unknown* | 15 | 3 | 0 | 5 | 7 | |
| antiTg_stat | | | | | | 0.880 |
| *Negative* | 3,334 (98%) | 576 (98%) | 119 (99%) | 1,575 (98%) | 1,064 (98%) | |
| *Positive* | 59 (1.7%) | 10 (1.7%) | 1 (0.8%) | 31 (1.9%) | 17 (1.6%) | |
| *Unknown* | 17 | 4 | 1 | 7 | 5 | |
| RIDRETH2 | | | | | | **<0.001** |
| *NH White* | 1,500 (44%) | 270 (46%) | 37 (31%) | 750 (46%) | 443 (41%) | |
| *NH Black* | 681 (20%) | 106 (18%) | 45 (37%) | 261 (16%) | 269 (25%) | |
| *Mexican Hispanic* | 582 (17%) | 91 (15%) | 19 (16%) | 263 (16%) | 209 (19%) | |
| *Other Hispanic* | 407 (12%) | 62 (11%) | 14 (12%) | 205 (13%) | 126 (12%) | |
| *Other Race* | 240 (7.0%) | 61 (10%) | 6 (5.0%) | 134 (8.3%) | 39 (3.6%) | |
| LBXSAL | 4.20 (4.00, 4.40) | 4.30 (4.10, 4.50) | 4.10 (4.00, 4.30) | 4.30 (4.10, 4.50) | 4.10 (3.90, 4.30) | **<0.001** |
| LBXSGTSI | 21 (15, 32) | 16 (12, 23) | 18 (14, 27) | 21 (15, 32) | 25 (18, 36) | **<0.001** |
| *Unknown* | 1 | 0 | 0 | 1 | 0 | |
| LBXSTB | 0.70 (0.60, 0.90) | 0.80 (0.60, 1.00) | 0.70 (0.50, 0.90) | 0.80 (0.60, 1.00) | 0.70 (0.60, 0.90) | **<0.001** |
| *Unknown* | 2 | 0 | 0 | 2 | 0 | |
| LBXSUA | 5.50 (4.60, 6.50) | 4.70 (3.90, 5.60) | 5.40 (4.40, 6.50) | 5.40 (4.60, 6.40) | 6.00 (5.00, 6.90) | **<0.001** |
| *Unknown* | 1 | 0 | 0 | 1 | 0 | |
| LBXSGB | 2.90 (2.60, 3.20) | 2.80 (2.60, 3.10) | 3.00 (2.80, 3.30) | 2.90 (2.60, 3.20) | 3.00 (2.70, 3.30) | **<0.001** |
| *Unknown* | 4 | 0 | 0 | 2 | 2 | |
| FT4_T3_Ratio | 2.49 (2.19, 2.77) | 2.50 (2.22, 2.74) | 2.43 (2.12, 2.66) | 2.49 (2.19, 2.81) | 2.46 (2.19, 2.76) | 0.227 |
| *Unknown* | 3 | 0 | 0 | 1 | 2 | |
| TT4_T3_Ratio | 68 (60, 78) | 67 (59, 76) | 66 (58, 74) | 68 (60, 79) | 69 (61, 79) | **<0.001** |
| *Unknown* | 15 | 5 | 1 | 4 | 5 | |

[1]n (%); Median (IQR)

[2]Fisher's Exact Test for Count Data with simulated p-value (based on 2000 replicates); Kruskal-Wallis rank sum test

Additionally, smoking status (smok_stat2) demonstrated a significant difference ($p < 0.001$) among the groups. The majority of individuals in all groups were non-smokers, but a higher proportion of current smokers was observed in the MUNW and MUO groups compared to the other groups.

No significant differences were found for TPO_stat ($p = 0.928$) and antiTg_stat ($p = 0.880$), indicating similar distributions among the groups. Ethnicity (RIDRETH2) displayed a significant difference ($p < 0.001$), with variations in the proportions of different racial and ethnic groups across the MHNW, MHO, MUNW, MUO, and unknown groups.

Furthermore, several laboratory measurements showed significant differences among the groups. These included serum albumin (LBXSAL, $p < 0.001$), serum globulin (LBXSGTSI, $p < 0.001$), total bilirubin (LBXSTB, $p < 0.001$), serum uric acid (LBXSUA, $p < 0.001$), and serum globulin (LBXSGB, $p < 0.001$). The FT4_T3_Ratio ($p = 0.227$) and TT4_T3_Ratio ($p < 0.001$) did not show significant differences among the groups.

**Table 4. Final model performance.**

| Model | Hand-Till AUROC | Accuracy | Precision | Recall | F1 |
|---|---|---|---|---|---|
| Multinomial Logistic Regression | 0.818 | 0.655 | 0.559 | 0.502 | 0.510 |
| Neural Network | 0.814 | 0.655 | 0.632 | 0.487 | 0.640 |
| Random Forest | 0.811 | 0.641 | 0.647 | 0.439 | 0.599 |
| Boosted Trees | 0.811 | 0.648 | 0.621 | 0.458 | 0.612 |
| K Nearest Neighbors | 0.786 | 0.604 | 0.621 | 0.379 | 0.512 |
| Lasso Regression | 0.776 | 0.633 | 0.613 | 0.435 | 0.587 |

We employed multinomial logistic regression to analyze the associations between the metabolic phenotypes and the independent variables presented in the demographic table. Additionally, we utilized machine learning approaches, including neural network, random forest, boosted trees, lasso regression, and K nearest neighbors models, to explore and validate the relationships between the variables. The inclusion of these machine learning techniques allowed for a comprehensive evaluation of the predictive performance and potential insights gained from alternative modeling approaches. We evaluated the performance of each model based on metrics such as Hand-Till AUROC, accuracy, precision, recall, and F1 score (Table 4).

A confusion matrix that shows the distribution of test predictions against their true values is presented in Fig 3. A gradient is overlayed onto the numeric values to produce a heatmap. It is clear that there is a class imbalance with MUNW being the largest class and MHO being the smallest class. The only model to predict the minority class is the multinomial logistic regression.

## Multinomial logistic regression

The multinomial logistic regression model achieved the best performance with an AUROC of 0.818, indicating its ability to discriminate between classes. The model exhibited an accuracy of 0.655, precision of 0.559, recall of 0.502, and an F1 score of 0.510. These results suggest that the model performed moderately well in predicting the target variable.

## Neural network

The neural network model yielded an AUROC of 0.814, demonstrating its discriminative power. The model achieved an accuracy of 0.655, precision of 0.632, recall of 0.487, and an F1 score of 0.640. These metrics suggest that the neural network performed similarly to the multinomial logistic regression model, showing competitive performance in predicting the target variable.

## Random forest

The random forest model produced an AUROC of 0.811, indicating its ability to discriminate between classes. The model exhibited an accuracy of 0.641, precision of 0.647, recall of 0.439, and an F1 score of 0.599. These results suggest that the Random Forest model achieved reasonable predictive performance, albeit with a lower recall compared to the previous models. The most important predictors based on purity was HOMAQ_Q4 with an importance score of 88.2, followed closely by RIDAGEYR with an importance score of 84.8 (Fig 1). Other important predictors included LBXSUA (56.4), LBXSAL (49.7), and LBXSGTSI (48.5). Additionally, TT4_T3_Ratio (36.7) and FT4_T3_Ratio (32.3) were also notable contributors to the model.

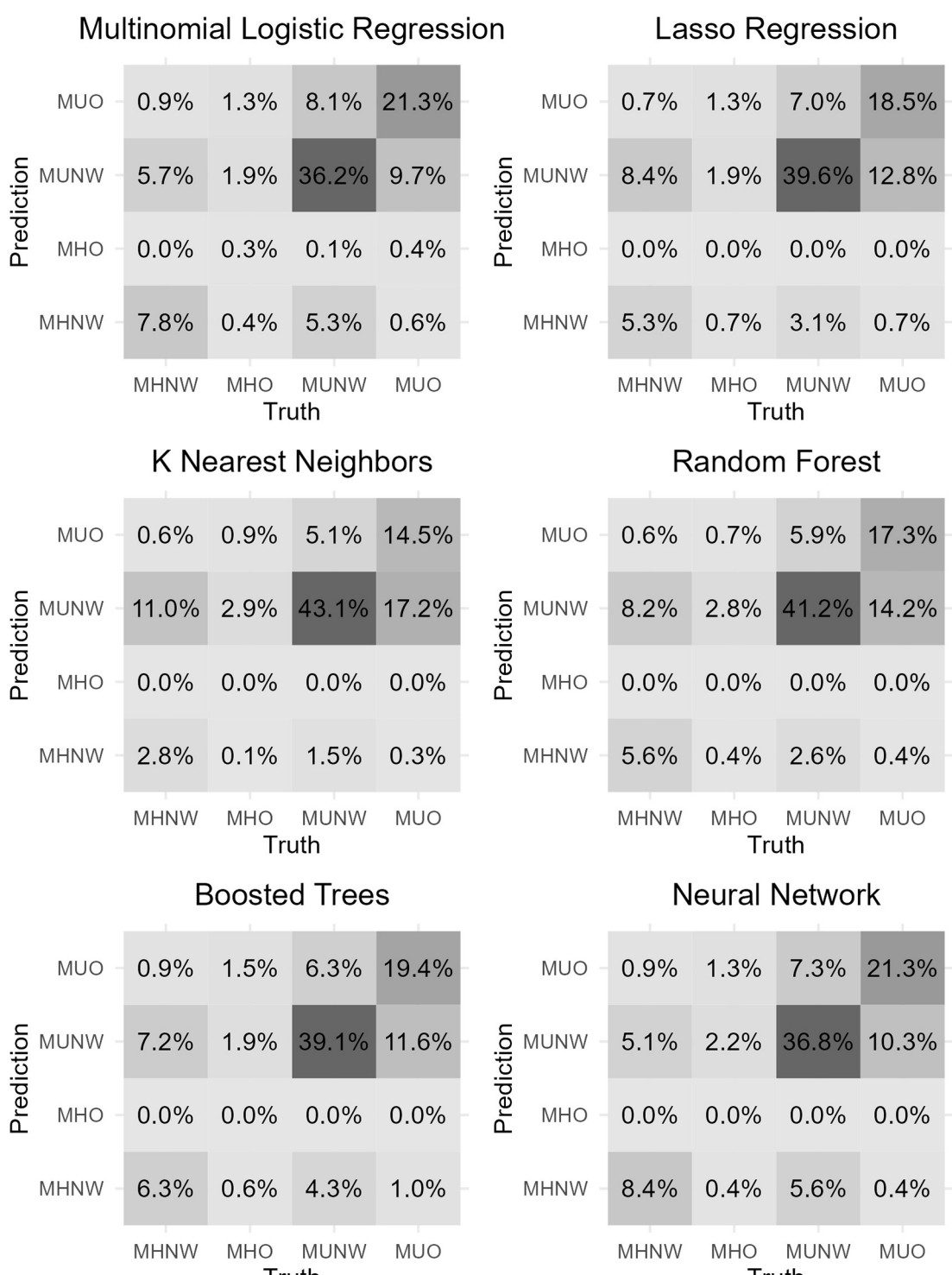

**Fig 3. Test prediction confusion matrix.**

Among the demographic variables, RIAGENDR_Female (10.3) showed some influence, as did smok_stat2_Current.Smoker (7.5) and RIDRETH2_NH.Black (6.0).

### Boosted trees

The boosted trees model also achieved an AUROC of 0.811, demonstrating its discriminative power. The model yielded an accuracy of 0.648, precision of 0.621, recall of 0.458, and an F1 score of 0.612. These results indicate that the Boosted Trees model performed similarly to the Random Forest model, with comparable predictive performance.

### K nearest neighbors

The K nearest neighbors model obtained an AUROC of 0.786, indicating its discriminative ability. The model exhibited an accuracy of 0.604, precision of 0.621, recall of 0.379, and an F1 score of 0.512. These results suggest that the K nearest neighbors model achieved moderate predictive performance, but with a lower recall compared to the other models.

### Lasso regression

The lasso regression model had the lowest AUROC with a value of 0.776. Although it was presented many interaction effects, most of the predictors were eliminated during regularization in the tuning stage. The model achieved an accuracy of 0.633, precision of 0.613, recall of 0.435, and an F1 score of 0.587. Despite its low AUROC curve, the model's F1 score shows a better balance of precision and recall than the K nearest neighbors, random forest, and multinomial logistic regression models.

In summary, the multinomial logistic regression, neural network, random forest, boosted trees, and K nearest neighbors models were evaluated in terms of their performance metrics. The neural network and multinomial logistic regression models performed relatively well, achieving similar AUROC scores and demonstrating competitive accuracy, precision, recall, and F1 scores. The random forest and boosted trees models exhibited slightly lower AUROC scores but maintained reasonable predictive performance. Finally, the K nearest neighbors model achieved moderate performance, with lower recall compared to the other models.

## Discussion

Our study reveals distinctive characteristics in the predictive performance of each machine learning model for metabolic phenotypes, as evident from their performance metrics. Both the Multinomial Logistic Regression and Neural Network models demonstrate moderate overall performance (AUROC of 0.818 and 0.814, respectively), with the former tending towards a balanced precision and recall, while the latter tends to prioritize precision over recall. In contrast, both the Random Forest and Boosted Trees methods exhibit competitive AUROC scores, underscoring their ability to capture underlying data patterns effectively. However, while the Random Forest model demonstrates higher precision, Boosted Trees display a more balanced precision-recall trade-off. Conversely, the K Nearest Neighbors model struggles with lower recall, indicating limitations in effectively identifying positive cases. Lastly, the Lasso Regression model, while maintaining decent precision, falls short in recall, suggesting potential challenges in capturing all instances of positive metabolic phenotypes.

In this study, parameters possibly related to both thyroid function and obesity were selected for ML variables. It is important to note that the performance of machine learning models can vary depending on the specific dataset, the quality and quantity of data, the chosen evaluation metrics, and other factors. Therefore, while Multinomial Logistic Regression performed best

in this study, it does not imply that it will always be the top-performing model in every scenario. It is advisable to assess the suitability of different models based on the specific requirements and characteristics of the dataset at hand. Machine learning performs better in scenarios where large and complex datasets with numerous variables and intricate patterns need to be analyzed, as they excel at extracting valuable insights and making accurate predictions. They are well-suited for capturing non-linear relationships between variables, handling high-dimensional data by automatically learning relevant features or performing feature selection. Machine learning techniques are also capable of processing unstructured data such as text, images, audio, and video, enabling meaningful information extraction and classification. They can be utilized for real-time decision making, large-scale data processing, and prediction/classification tasks, demonstrating their versatility in various domains. However, the performance of machine learning algorithms depends on factors like data quality, feature engineering, model selection, hyperparameter tuning, and the expertise of the practitioner, necessitating careful assessment of specific requirements to determine the suitability of machine learning approaches [34].

When working with survey data such as NHANES, it is crucial to account for the complex survey design and ensure proper representation of the population. NHANES provides sample weights that allow researchers to estimate population-level statistics accurately. These weights adjust for sampling probabilities, non-response, and post-stratification factors. Failure to incorporate sample weights can lead to biased estimates and incorrect inferences. MacNell et al. [35] conducted a study examining the implications of incorporating sampling weights in gradient boosting models. Their findings emphasized the importance of accounting for sampling weights when analyzing data from complex surveys. The authors concluded that neglecting to consider sampling weights in gradient boosting models can potentially compromise the generalizability of the results, particularly depending on the sample size and other analytical characteristics. In cases where weighted algorithms are not readily available, the study suggests performing post-hoc re-calculations of unweighted model performance using weighted observed outcomes as an alternative approach. This methodology may provide a more accurate reflection of the model's predictions in target populations than completely disregarding the weights. The study by MacNell et al. underscores the significance of appropriately handling sampling weights to ensure valid and reliable analyses in complex survey settings.

Expanding the set of variables in the analysis can enhance the understanding and predictive power of the models. NHANES provides a rich array of health-related variables that capture various aspects of individuals' health status, behaviors, and demographic characteristics. By incorporating additional variables, researchers can potentially uncover important relationships, identify new risk factors, or gain insights into complex interactions. However, it is crucial to consider the relevance and validity of the additional variables and ensure they align with the research objectives and hypotheses for machine learning approaches. Careful variable selection and feature engineering techniques may be necessary to prevent issues such as multi-collinearity and overfitting. Adding more variables can improve the models' accuracy and robustness, but it requires thoughtful consideration and domain expertise. Another limitation of our study is that although using NHANES data from different cycles for validation could potentially enhance the analytical value, a significant number of parameters used in our research were not included in data from other cycles. This discrepancy limited our ability to conduct a comprehensive validation. Furthermore, it's essential to acknowledge that while our findings are based on NHANES data, their generalizability to other datasets may be limited. Therefore, additional research is warranted to explore these relationships in different populations and settings, which could inform future investigations in this field. Lastly, there is a large class imbalance in the outcome variable, especially between MUNW (the majority class) and

MHO (the minority class). This results in a scarcity of MHO predictions from the models. To counteract this, future work could incorporate the synthetic minority over-sampling technique (SMOTE) [36] to make sure that all outcome classes are represented in the predictions. SMOTE creates observations from the feature space of the minority class so that the synthetic data is within the realm of what is reasonable but not redundant.

Future studies should also explore the integration of machine learning techniques to address the challenges associated with survey sample weights and the incorporation of additional variables. Machine learning algorithms have the potential to handle complex survey data and leverage the power of sample weights effectively. Researchers can investigate novel approaches that specifically account for sample weights within machine learning models, ensuring accurate and representative analysis of survey data. A follow-up study could create Neural nets with more complex architecture than the one used in this study (multilayer perceptron) to model sophisticated relationships between predictors. Additionally, exploring the impact of incorporating more variables in machine learning models can uncover hidden patterns and relationships, enhancing prediction accuracy and providing a deeper understanding of the factors influencing the outcomes of interest. Future research in this domain can pave the way for innovative methodologies that combine machine learning, sample weights, and expanded variable sets, ultimately advancing the field of survey data analysis and enabling more robust and reliable insights.

## Author Contributions

**Conceptualization:** Hyeong Jun Ahn, Min-Hee Kim.

**Data curation:** Kyle Ishikawa.

**Formal analysis:** Kyle Ishikawa.

**Investigation:** Hyeong Jun Ahn.

**Methodology:** Hyeong Jun Ahn, Kyle Ishikawa.

**Supervision:** Hyeong Jun Ahn.

**Writing – original draft:** Hyeong Jun Ahn, Min-Hee Kim.

**Writing – review & editing:** Hyeong Jun Ahn, Kyle Ishikawa, Min-Hee Kim.

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
