## [Decision Letter · Decision Letter 0]

13 Feb 2024

PONE-D-23-34518Exploring the diagnostic performance of machine learning in prediction of metabolic phenotypes focusing on thyroid functionPLOS ONE

Dear Dr. Ahn,

Thank you for submitting your manuscript to PLOS ONE. After careful consideration, we feel that it has merit but does not fully meet PLOS ONE’s publication criteria as it currently stands. Therefore, we invite you to submit a revised version of the manuscript that addresses the points raised during the review process.

We look forward to receiving your revised manuscript.

Kind regards,

Vijayalakshmi Kakulapati, Ph.D

Academic Editor

PLOS ONE

Journal Requirements:

[ Hyeong Jun Ahn and Kyle Ishikawa are partially supported by the National Institute of Health (2U54MD007601-36 and U54GM138062). The content is solely the responsibility of the authors and does not necessarily represent the official views of the NIH.]

 [The author(s) received no specific funding for this work.]

5. Please include a copy of Table 3 which you refer to in your text on page 18.

Reviewers' comments:

Reviewer's Responses to Questions

**Comments to the Author**

1. Is the manuscript technically sound, and do the data support the conclusions?

Reviewer #1: Yes

Reviewer #2: Partly

Reviewer #3: Yes

2. Has the statistical analysis been performed appropriately and rigorously? 

Reviewer #1: Yes

Reviewer #2: Yes

Reviewer #3: Yes

3. Have the authors made all data underlying the findings in their manuscript fully available?

Reviewer #1: Yes

Reviewer #2: Yes

Reviewer #3: Yes

4. Is the manuscript presented in an intelligible fashion and written in standard English?

Reviewer #1: Yes

Reviewer #2: Yes

Reviewer #3: Yes

5. Review Comments to the Author

Reviewer #1: 1- The name and information about the database used were not mentioned in the abstract

2- In the introduction, please add the researcher’s contributions. In addition to adding a paragraph at the end of the introduction explaining the structure of the research

3- The working algorithm is not clear at all. In addition, the reasons for choosing them or the difference between them were not explained. Also, please pay attention to the numbering of titles

4-The results for the metrics are presented in a table rather than in text form

5- The discussion is devoid of any mention of the results with their numerical values. It is also preferable to add the conclusions in a separate section

6-Why is deep learning not mentioned as future work? Is it unlikely for certain reasons?

7-References need to be more standardized and organized

Reviewer #2: 1. It only used data from a single study/dataset (NHANES), limiting generalizability. Using multiple datasets could strengthen findings.

2. Performance comparisons were limited to 5 commonly used algorithms. Testing more recent deep learning methods may yield better results.

3. Validating models on held-out test data from a later NHANES wave would better reflect real-world use, versus internal validation.

4. Limited demographic/clinical covariates were considered; including more exhaustive potential predictors could impact results.

5. Interactions and non-linear effects between variables were not modeled, which may be important for complex phenotypes.

6. Describe dataset features in more details and its total size and size of (train/test) as a table.

7. Pseudocode / Flowchart and algorithm steps need to be inserted.

8. Time spent need to be measured in the experimental results.

9. Limitation Section need to be inserted.

10. The architecture of the proposed model must be provided

11. Address the accuracy/improvement percentages in the abstract and in the conclusion sections, as well as the significance of these results.

12. The authors need to make a clear proofread to avoid grammatical mistakes and typo errors.

13. Add future work in last section (conclusion) (if any)

14. To improve the Related Work and Introduction sections authors are recommended to review this highly related research work paper:

a) Optimizing epileptic seizure recognition performance with feature scaling and dropout layers

b) Optimizing classification of diseases through language model analysis of symptoms

c) Predicting female pelvic tilt and lumbar angle using machine learning in case of urinary incontinence and sexual dysfunction

d) Utilizing convolutional neural networks to classify monkeypox skin lesions

e) Hepatitis C Virus prediction based on machine learning framework: a real-world case study in Egypt

Reviewer #3: Thank you for the hard work presented. It is nicely written and covers an important topic.

Authors could take the advantages of ML algorithms in modeling one of the medical diagnosis applications. However, could provide them with some few comments:

1- Generally, along the sections, some paragraphs with the same information have been duplicated. Try to remove them as much as possible.

2- Authors could provide a section describing the dataset, its complexity (if) and other details that have direct impact on the outcome of the ML models.

3- The paragraph from the line 150 - 166, perhaps this paragraph be moved earlier when authors talk about preprocessing and Table 1.

4- in line 152, there is Table X, is it the Table 1 or another table not being included in the document?

5- Why did not the authors provide a Figure shows the AUROC?

6- Authors could compared their results with some state-of-art works?

7- lines 258 - 260: this is a general advantage of ML that already being presented in the Introduction. Have you developed a real-time ML model?

8- If you believe that the dataset you are using is complex, why you used a regular Neural networks? why you did not implemented the deep learning?

9- How can we reflect the work of MacNell on yours? How can you use their conclusions to support your results?

10- You have mentioned about developing a confusion matrix. Can you show a Figure visualizing them?

6. PLOS authors have the option to publish the peer review history of their article (what does this mean?). If published, this will include your full peer review and any attached files.

Reviewer #1: No

Reviewer #2: **Yes: **Tarek Abd El-Hafeez

Reviewer #3: No

---

## [Author Response · Author response to Decision Letter 0]

19 Apr 2024

Point by point response to reviewer’s comments

Note: We have provided our response in blue. The line and page numbers referenced correspond to the tracking version of the manuscript.

Reviewer #1: 1- The name and information about the database used were not mentioned in the abstract

Thank you for your comment. We utilized the National Health and Nutrition Examination Survey (NHANES) from 2007 to 2012 and made a correction in Abstract. 

“In this study, we employed various machine learning models to predict metabolic phenotypes, focusing on thyroid function, using a dataset from the National Health and Nutrition Examination Survey (NHANES) from 2007 to 2012. ”

2- In the introduction, please add the researcher’s contributions. In addition to adding a paragraph at the end of the introduction explaining the structure of the research

We have included the following information on line 91 of page 5. “Hyeong Jun Ahn, as the first author, spearheads the study's conceptualization, protocol development, and manuscript drafting, collaborating with co-authors to refine the submission. Additionally, Minhee Kim provides valuable clinical expertise through an exhaustive literature review, synthesizing existing knowledge to inform the study's background and rationale. Furthermore, Kyle Ishikawa plays a pivotal role in the research endeavor, spearheading all aspects of data management, organization, and analysis, ensuring the integrity and accuracy of the study's findings.”

3- The working algorithm is not clear at all. In addition, the reasons for choosing them or the difference between them were not explained. Also, please pay attention to the numbering of titles

Details and strengths about each algorithm have been added to the methods section where the finalized hyperparameters are discussed. Also, a flowchart of the general ML steps for each algorithm is presented as Figure 2. 

4-The results for the metrics are presented in a table rather than in text form

In consideration of the reviewer's valuable feedback, the decision was made to remove redundant metrics mentioned in the text, as it was deemed that the performance table offers a more precise presentation of the data.

5- The discussion is devoid of any mention of the results with their numerical values. It is also preferable to add the conclusions in a separate section

As the reviewer pointed out, we have incorporated a summary of important numeric values (performance of each machine learning algorithm) from the results into the discussion section as follows (page 23, line 439).

Our study reveals distinctive characteristics in the predictive performance of each machine learning model for metabolic phenotypes, as evident from their performance metrics. Both the Multinomial Logistic Regression and Neural Network models demonstrate moderate overall performance (AUROC of 0.818 and 0.814, respectively), with the former tending towards a balanced precision and recall, while the latter tends to prioritize precision over recall. In contrast, both the Random Forest and Boosted Trees methods exhibit competitive AUROC scores, underscoring their ability to capture underlying data patterns effectively. However, while the Random Forest model demonstrates higher precision, Boosted Trees display a more balanced precision-recall trade-off. Conversely, the K Nearest Neighbors model struggles with lower recall, indicating limitations in effectively identifying positive cases. Lastly, the Lasso Regression model, while maintaining decent precision, falls short in recall, suggesting potential challenges in capturing all instances of positive metabolic phenotypes.

6-Why is deep learning not mentioned as future work? Is it unlikely for certain reasons?

Thank you for mentioning this. Our final model is a multilayer perceptron (MLP) which is a feed-forward neural network. Future work can be done to compare different neural net architectures and their suitability for our data.

7-References need to be more standardized and organized

Thank you for providing feedback. We have carefully reviewed the references and standardized them to comply with the Vancouver style

Reviewer #2: 1. It only used data from a single study/dataset (NHANES), limiting generalizability. Using multiple datasets could strengthen findings.

Thank you for your advice. We added your feedback as limitation (page 26, line 502)

2. Performance comparisons were limited to 5 commonly used algorithms. Testing more recent deep learning methods may yield better results.

Thank you for this suggestion. Future work can include more complicated neural net architectures that will address our dataset better.

3. Validating models on held-out test data from a later NHANES wave would better reflect real-world use, versus internal validation.

We agree with and appreciate the reviewer's comments. As pointed out, using later NHANES data for validation could indeed enhance the value of the validation process. However, since NHANES data is collected in cycles with varying parameters, the parameters used in this study between 2007 and 2012 are not consistently collected in subsequent cycles, leading to a mix of available and unavailable parameters. This presents an inherent limitation in conducting validation with later cycles. Therefore, we believe it is reasonable to mention this as a limitation of our study (page 26, line 498)

Correction in Discussion 

"Another limitation of our study is that although using NHANES data from different cycles for validation could potentially enhance the analytical value, a significant number of parameters used in our research were not included in data from other cycles. This discrepancy limited our ability to conduct a comprehensive validation."

4. Limited demographic/clinical covariates were considered; including more exhaustive potential predictors could impact results.

We appreciate the reviewer's comment. We initially believed that applying as many parameters as possible (exhaustive potential predictors) to our analysis model would aid in improving outcomes across various machine learning scenarios. However, the actual addition of these parameters did not yield a significant improvement in results. This lack of significant enhancement may be attributed to the fact that in our study, we selectively applied parameters already associated with metabolic phenotype. 

5. Interactions and non-linear effects between variables were not modeled, which may be important for complex phenotypes.

Thank you for the suggestion. We have added a lasso regression model to the project for variable selection of the interaction terms. In the preprocessing phase for the lasso model, every two- and three-way combination of interaction terms were created for the lasso model to reduce.

6. Describe dataset features in more details and its total size and size of (train/test) as a table.

Thank you for the comment. An overall summary table for our complete dataset has been added as Table 3, and details about the training-test split has been added to the flowchart (next item).

7. Pseudocode / Flowchart and algorithm steps need to be inserted.

A flowchart for the general ML steps has been added to the end of the methods section.

8. Time spent need to be measured in the experimental results.

Time to train the tuning models was added to Table 2.

9. Limitation Section need to be inserted.

Limitation was inserted in line 489-513 at page 26. 

10. The architecture of the proposed model must be provided

The architecture of the final neural net has been described in the methods section and Figure 2 was created to show the architecture.

11. Address the accuracy/improvement percentages in the abstract and in the conclusion sections, as well as the significance of these results.

Since we focus on the discriminating power we’ve emphasized AUROC in the abstract. AUROC quantifies the model's ability to discriminate between the positive and negative classes across all possible thresholds. It represents the probability that the model will rank a randomly chosen positive instance higher than a randomly chosen negative instance. 

12. The authors need to make a clear proofread to avoid grammatical mistakes and typo errors.

Thank you for your comments. We made a proofread again. 

13. Add future work in last section (conclusion) (if any)

We added a sentence about deep learning with more complex architectures in the final paragraph (page 27, line 519).

14. To improve the Related Work and Introduction sections authors are recommended to review this highly related research work paper:

We have incorporated the content of the papers suggested by the reviewer into the introduction, relating to the expansion of machine learning applications within the healthcare sector (page 4, line 58).

a) Optimizing epileptic seizure recognition performance with feature scaling and dropout layers

b) Optimizing classification of diseases through language model analysis of symptoms

c) Predicting female pelvic tilt and lumbar angle using machine learning in case of urinary incontinence and sexual dysfunction

d) Utilizing convolutional neural networks to classify monkeypox skin lesions

e) Hepatitis C Virus prediction based on machine learning framework: a real-world case study in Egypt

Reviewer #3: Thank you for the hard work presented. It is nicely written and covers an important topic.

Authors could take the advantages of ML algorithms in modeling one of the medical diagnosis applications. However, could provide them with some few comments:

1- Generally, along the sections, some paragraphs with the same information have been duplicated. Try to remove them as much as possible.

Thank you for the feedback. As we addressed Reviewer 1’s feedback, we decided to remove redundant metrics mentioned in the text, as it was deemed that the performance table offers a more precise presentation of the data.

2- Authors could provide a section describing the dataset, its complexity (if) and other details that have direct impact on the outcome of the ML models.

We addressed the complexity of NHANES data (page 25, line 473) and the impact of ML models (page 24, line 439). 

3- The paragraph from the line 150 - 166, perhaps this paragraph be moved earlier when authors talk about preprocessing and Table 1.

Thank you for this suggestion. We agree, and this paragraph has been combined with the paragraph before Table 1.

4- in line 152, there is Table X, is it the Table 1 or another table not being included in the document?

Thanks for catching this. We removed this portion of the paragraph due to combining it with the paragraph before Table 1.

5- Why did not the authors provide a Figure shows the AUROC?

While figures can be useful for visualizing certain aspects of data, such as trends or patterns, we believed that a table format would offer a more concise and straightforward presentation of the AUROC and other accuray values for each model. By organizing the outcome values in a table, readers can readily compare the performance of the different models across multiple metrics in a structured and easily interpretable format. This approach not only streamlines the presentation of results but also allows for a more efficient utilization of space within the manuscript. Furthermore, tables are particularly advantageous when presenting numerical data with a high degree of precision, as they provide a clear delineation of values without the potential distortion that can occur in graphical representations. By prioritizing clarity and precision in the presentation of AUROC and other accuracy values, we aimed to ensure that readers could accurately assess the performance of each model and make informed comparisons between them.

6- Authors could compared their results with some state-of-art works?

We acknowledge the importance of contextualizing our findings within the broader landscape of existing research and it's essential to highlight that the comparison of our results with state-of-the-art works in the field of metabolic phenotypes for thyroid function was not feasible due to the lack of directly comparable studies in the current literature. The unique focus of our investigation on metabolic phenotypes specifically related to thyroid function presents a novel contribution to the scientific discourse, with limited existing studies directly addressing this intersection.

7- lines 258 - 260: this is a general advantage of ML that already being presented in the Introduction. Have you developed a real-time ML model?

We have not created an interface where researchers can input patient characteristics. However, in our ML pipeline, we can produce outcome predictions if we have a structured dataset of predictors. Perhaps future work can look into creating a web dashboard for researchers to use.

8- If you believe that the dataset you are using is complex, why you used a regular Neural networks? why you did not implemented the deep learning?

The single-hidden layer neural net did not perform better than the logistic regression, so we did not feel like we needed to explore more complicated architectures. However, that could be a point of future work.

9- How can we reflect the work of MacNell on yours? How can you use their conclusions to support your results?

Actually we tried the sampling weights on our models but we could not find a significant difference. We believe that the impact will be present when outcome variables are significantly associated with the sampling weights. 

10- You have mentioned about developing a confusion matrix. Can you show a Figure visualizing them?

Yes, thanks for catching that. A confusion matrix for each model has been added to the results section after the performance table.

---

## [Decision Letter · Decision Letter 1]

20 May 2024

Exploring the diagnostic performance of machine learning in prediction of metabolic phenotypes focusing on thyroid function

PONE-D-23-34518R1

Dear Dr. Ahn,

We’re pleased to inform you that your manuscript has been judged scientifically suitable for publication and will be formally accepted for publication once it meets all outstanding technical requirements.

Kind regards,

Vijayalakshmi Kakulapati, Ph.D

Academic Editor

PLOS ONE

**Comments to the Author**

Reviewer #1: 1- The abstract needs a general beginning before going into the details of the proposed work.

2- Adding a paragraph at the end of the introduction that explains the structure of the research presented in all its sections

3- What does this mean on page 6: (supplemental table **).

4-The proposed model is not clear, and it is preferable to add a general structure or flow chart that explains the stages of work

5-Add at least two references published in the year 2024

Reviewer #2: An updated manuscript addressing previous comments and suggestions was evaluated positively. The updated submission demonstrates significant improvement and provides valuable insights relevant to the research community.

Reviewer #3: The Authors have addressed the comments. In addition, within the comments of other reviewers, now the manuscript becomes more conducted and deserve to go for publication.

---

## [Editor Report · Acceptance letter]

30 May 2024

PONE-D-23-34518R1 

PLOS ONE

Dear Dr. Ahn, 

I'm pleased to inform you that your manuscript has been deemed suitable for publication in PLOS ONE. Congratulations! Your manuscript is now being handed over to our production team.

Kind regards, 

on behalf of

Dr. Vijayalakshmi Kakulapati 

Academic Editor

PLOS ONE